# Effects of Intense Laser Field on Electronic and Optical Properties of Harmonic and Variable Degree Anharmonic Oscillators

**DOI:** 10.3390/nano12101620

**Published:** 2022-05-10

**Authors:** Melike Behiye Yücel, Esin Kasapoglu, Carlos A. Duque

**Affiliations:** 1Department of Physics, Faculty of Science, Akdeniz University, Antalya 07058, Turkey; 2Department of Physics, Faculty of Science, Sivas Cumhuriyet University, Sivas 58140, Turkey; ekasap@cumhuriyet.edu.tr; 3Grupo de Materia Condensada-UdeA, Instituto de Física, Facultad de Ciencias Exactas y Naturales, Universidad de Antioquia UdeA, Calle 70 No. 52-21, Medellín, Colombia; carlos.duque1@udea.edu.co

**Keywords:** harmonic oscillator, anharmonic oscillator, absorption spectrum, intense laser field

## Abstract

In this paper, we calculated the electronic and optical properties of the harmonic oscillator and single and double anharmonic oscillators, including higher-order anharmonic terms such as the quartic and sextic under the non-resonant intense laser field. Calculations are made within the effective mass and parabolic band approximations. We have used the diagonalization method by choosing a wave function based on the trigonometric orthonormal functions to find eigenvalues and eigenfunctions of the electron confined within the harmonic and anharmonic oscillator potentials under the non-resonant intense laser field. A two-level approach in the density matrix expansion is used to calculate the linear and third-order nonlinear optical absorption coefficients. Our results show that the electronic and optical properties of the structures we focus on can be adjusted to obtain a suitable response to specific studies or aims by changing the structural parameters such as width, depth, coupling between the wells, and applied field intensity.

## 1. Introduction

The harmonic oscillator (HO) potential is one of the most important model potentials in quantum mechanics used to describe a molecular vibration in the very close neighborhood of a stable equilibrium point and one of the few quantum-mechanical systems with exact and analytical solutions. Adding anharmonic oscillator (AHO) potentials to the HO potential better describes molecular vibrations, certainly. The AHO potential defines an oscillator that is not oscillating in simple harmonic motion. Namely, the restoring force is no longer proportional to the displacement. Adding the higher-order anharmonic terms such as the cubic, quartic, and sextic to the HO further improves the HO approximation, especially under greater displacement from the equilibrium position.

The transport of heat along a chain of particles interacting through anharmonic potentials consisting of quartic terms in addition to harmonic quadratic terms and subject to heat reservoirs at its ends have been reported by Landi and Oliveira, showing that the introduction of the energy conserving stochastic noise leads to Fourier’s law [1]. In the theoretical work of N. H. Fletcher, it is concluded that the term “inharmonic” is an appropriate descriptor for classical oscillators, such as metal bars, that have nonharmonic vibrational spectra even in the linear limit of small vibrations [2]. By using an exact and perturbative form, the repulsive harmonic oscillator and an extension of this system, with an additional linear anharmonicity term, have been studied [3]. The authors compared the perturbative solution up to second order with the exact solution when the system is initially in coherent and a Schrödinger-cat states. Panek and co-workers proposed an algorithm for the localization of normal modes in suitable subsets that are chosen to strictly limit the errors introduced by the harmonic couplings while still leading to maximally localized modes [4]. By a suitable choice of coordinates and by using their proposed algorithm, the computational effort required for calculations of anharmonic vibrational spectra can be reduced significantly. Sharma and Sastri reported the numerical solution of the quantum anharmonic oscillator by using the matrix diagonalization technique [5]. In their work, where the interaction potential consists of quadratic and quartic terms, it was embedded within an infinite square well potential of appropriate width, and its sine eigenfunctions were used as basis functions for the employed diagonalization matrix method.

The AHO is one of the most important problems, which cannot be solved analytically in contrast to the HO. So, it has been extensively studied both perturbatively and non-perturbatively [6,7,8,9,10,11,12,13,14,15,16,17,18,19]. Bender and Wu [6] showed that the perturbation series for the AHO with a quartic anharmonicity term diverged for all values of the coupling parameter (λ) by direct calculation. Most of the non-perturbative methods were used after these results. P. K. Patnaik [10] developed Rayleigh–Schrödinger (RS) perturbation theory, which consists of two steps for the AHO with a quartic anharmonicity term. The perturbation series is shown to be convergent. The theory allows for the correct limit in both large and small anharmonicity. Additionally, why standard RS perturbation theory does not work for an AHO is explained in detail in this paper. The AHO potentials with sextic, octic, etc., terms have also been studied extensively [20,21,22,23,24,25,26,27,28] since these potentials play an important role in the quantum tunneling time problems, and in the spectra of molecules. In this context, the electronic and optical properties of the HO and single and double anharmonic oscillators, including higher-order anharmonic terms such as the quartic and sextic under the non-resonant intense laser field, are investigated in this paper by changing the structural parameters and field intensity.

## 2. Theory

In the effective mass approximation, the Hamiltonian for the confined electron is given by
(1)$$H=\frac{{\vec{p}}^{2}}{2\;m^{*}}+V\left(x\right)\;,$$
where $\vec{p}$ is the momentum operator, $m^{*}$ is the electron effective mass, V(x) is the confinement potential that includes harmonic and anharmonic potentials of different order, and its functional form before the applied intense laser field (ILF) as follows
(2)$$V\left(x\right)=V_{0}\;\left[\lambda_{1}{\left(x/k\right)}^{2}+\lambda_{2}{\left(x/k\right)}^{4}+\lambda_{3}{\left(x/k\right)}^{6}\right]\;,$$
where, $V_{0}$ is depth of the quantum well, the *k*-parameter is related to the well width, and $\lambda_{1}$ is known as the coupling parameter, which is also related to the width of the well. As the $\lambda_{1}$-parameter increases the well width becomes narrow. Depending on the choice of the $\lambda_{2}$- and $\lambda_{3}$-parameters, the confinement potential turns into the single or double AHO potential. For $\lambda_{1}\neq 0$ and $\lambda_{2}=\lambda_{3}=0$ the potential is called HO, for $\lambda_{1}\neq 0$
$\left(\lambda_{1}<0\right)$, $\lambda_{2}\neq 0$
$\left(\lambda_{2}>0\right)$ and $\lambda_{3}=0$ the potential becomes single (double) quartic AHO potential, and for $\lambda_{1}\neq 0$
$\left(\lambda_{1}<0\right)$, $\lambda_{3}\neq 0$
$\left(\lambda_{3}>0\right)$, $\lambda_{2}=0$ the potential becomes single (double) sextic AHO potential.

In the presence of the *x*-linear polarized incident non-resonant ILF, the potential in Equation (Equation 2) turns into the form called laser-dressed potential [29,30,31,32], which is given by
(3)$$\langle V(x,\alpha_{0})\rangle =\frac{\Omega }{2\pi }\int_{0}^{2\pi /\Omega }V\left(x+\alpha_{0}\sin\Omega t\right)dt\;,$$
where $\alpha_{0}=eA_{0}/m^{*}\Omega$, $A_{0}$ and Ω are the laser-dressing or ILF parameter, the magnitude of the vector potential, and the angular frequency of the non-resonant intense laser field, respectively. The details of the equations given above for dressed potentials and the nonperturbative approach based on the Kramers–Henneberger translation transformation developed to describe the atomic behavior in intense high-frequency ILF can be found in Refs. [33,34]. By substituting Equation (Equation 3) into Equation (Equation 2), the bound-state energies of the electron are obtained from the numerical solution of the time-independent Schrödinger equation by using the diagonalization method [35]. Immediately after the energies and related wavefunctions are acquired, for transitions between any two electronic states, linear and non-linear absorption coefficients are found by using the perturbation expansion and the density matrix methods.

Using the relevant approaches mentioned before, expressions of the linear, third-order nonlinear, and total absorption coefficients (ACs) for the optical transitions are found as follows [31,32,33,34,35], respectively
(4)$$\beta^{\left(1\right)}\left(\omega \right)=\omega \sqrt{\frac{\mu_{0}}{\varepsilon_{r}}}\frac{|M_{ij}{|}^{2}\sigma_{\nu }\hbar \Gamma_{ij}}{{\left(\Delta E_{ji}-\hbar \omega \right)}^{2}+{\left(\hbar \Gamma_{ij}\right)}^{2}}\prime$$
(5)$$\beta^{\left(3\right)}\left(\omega ,I\right)=-2\omega \sqrt{\frac{\mu_{0}}{\varepsilon_{r}}}\;\;\frac{I}{n_{r}\;\varepsilon_{0}\;c}\;\;\frac{|M_{ij}{|}^{4}\sigma_{\nu }\hbar \Gamma_{ij}}{{\left[{\left(\Delta E_{ji}-\hbar \omega \right)}^{2}+{\left(\hbar \Gamma_{ij}\right)}^{2}\right]}^{2}}\;,$$
(6)$$\beta \left(\omega \right)=\beta^{\left(1\right)}\left(\omega \right)+\beta^{\left(3\right)}\left(\omega ,I\right)\;,$$
where, $\varepsilon_{r}=n_{r}^{2}\varepsilon_{0}$ is the real part of the permittivity, $\sigma_{\nu }$ is the carrier density in the system, $\mu_{0}$ is the vacuum permeability, $\Delta E_{ji}=E_{j}-E_{i}$ is the energy difference between two impurity states, $M_{ij}=\langle \psi_{i}\left|e\;x\right|\psi_{j}\rangle$, (i,j=1,2) is the transition matrix element between the eigenstates $\psi_{i}$ and $\psi_{j}$ for incident radiation polarized in the *x*-direction, $\Gamma_{ij}\left(=1/T_{ij}\right)$ is the relaxation rate, $T_{ij}$ is the inverse relaxation time, *c* is the speed of the light in free space, and *I* is the intensity of incident photon with the ω-angular frequency that leads to the intersubband optical transitions. It should be noted that $M_{jj}=M_{ii}=0$ due to the even symmetry of the confinement potential.

## 3. Results and Discussion

The parameter values in our numerical calculations are $\varepsilon =n_{r}^{2}=12.58$, $m^{*}=0.067m_{0}$ (where $m_{0}$ is the free electron mass), $V_{0}=228$ meV, $T_{12}=0.2$ ps, $\mu_{0}=4\pi \times 10^{-7}$${\mathrm{Hm}}^{-1}$, $\sigma_{\nu }=3.0\times 10^{22}$$m^{-3}$, and $I=5.0\times 10^{8}$${W/m}^{2}$. These parameters are suitable for GaAs/GaAlAs heterostructures [32].

In the absence and presence of the ILF, the HO potential and the squared wave functions of the first four energy levels in relevant energy values for k=5 nm and $\lambda_{1}=0.2$ as a function of *x*-growth direction coordinate are given in Figure 1a,b for $\alpha_{0}=0$ and $\alpha_{0}=5\;\mathrm{nm}$, respectively. As seen in both figures, the spacing of energies is equal; only the HO potential shifts towards higher energies with the effect of ILF.

In the absence $\left(\alpha_{0}=0\right)$, and the presence of the ILF $\left(\alpha_{0}=5\;\mathrm{nm}\right)$, the variations of the first four energy levels of the HO potential versus the width parameter-*k* for different $\lambda_{1}$ values are given in Figure 2a,b, respectively. The increase in the value of the *k*-parameter ($\lambda_{1}$-parameter) causes to increase (decrease) in the well widths and to decrease (increase) in all energies in the absence and the presence of the ILF. In the absence of the ILF, the energies of the excited levels increase in the form of consecutive odd multiples of the ground state energy such as $E_{2}=3\;E_{1}$, $E_{3}=5\;E_{1}$ and $E_{4}=7\;E_{1}$ for all considered $\lambda_{1}$-values. However, this behavior between the energies is not observed in the presence of ILF.

To see the changes better in the amplitudes and positions of the total ACs related to the intersubband transitions, the variations of the energy difference among the first four energy levels of an electron confined in the HO potential, and the squared reduced transition matrix element corresponding to the stated levels as a function of the width parameter are given in Figure 3a,b, respectively. It should be noted that the reduced transition matrix element equals $M_{ij}/e(=$$\eta_{ij})$. Furthermore, total ACs versus the resonance photon energy for different values of the laser-dressed parameter and $\lambda_{1}$-parameter is given in Figure 3c. The energy difference-$\Delta E_{ji}$ between any two levels is decreasing function of the well width, and it has the same value regardless of the ILF value. As $\lambda_{1}$ increases $\Delta E_{ji}$ increases just like energy values since the geometric confinement becomes stronger. The parameter $\eta_{ij}^{2}$ is an increasing function of the width parameter. For $\lambda_{1}=0.2$, the parameters $\eta_{ij}^{2}$ are smaller than those of $\lambda_{1}=0.5$. As seen in Figure 3c, for each $\lambda_{1}$ value, there is an absorption peak for all allowed transitions between the first four energy levels considered since the energy difference between any two levels is the same. Due to equal spacing of energy, all transitions occur at the same frequency, which is called single line spectrum [36]. For $\lambda_{1}=0.5$, the absorption peak position shifts towards the blue due to the increment in $\Delta E_{ji}$. The increase in the amplitudes of the peaks belonging to the transitions between the upper levels is quite high due to the great $\eta_{ij}^{2}$ parameter values.

In the absence ($\alpha_{0}=0$-solid lines) and presence of the ILF ($\alpha_{0}=5\;\mathrm{nm}$-dashed lines), the quartic single AHO potential, $V\left(x\right)=V_{0}\left[\lambda_{1}{\left(x/k\right)}^{2}+\lambda_{2}{\left(x/k\right)}^{4}\right]$, and squared wave function of the first four levels in relevant energy values versus the *x*-coordinate for k=5nm and $\lambda_{1}=\lambda_{2}=0.5$ values are given in Figure 4a. The quartic single AHO potential also narrows under the influence of ILF and shifts towards higher energies. Figure 4b,c are shown the variations of the first four energy levels as a function of the *k*-parameter for a constant $\lambda_{1}$ value and two different $\lambda_{2}$ values in the absence and presence of ILF, respectively. For quartic single AHO potential, the variation of energies according to the mentioned parameters are as in HO, but the energy levels are not equally spaced. In the strong confinement regime, while the energies are quite sensitive to $\lambda_{2}$ and $\alpha_{0}$ parameters, they are almost independent in weak confinement. In the absence and presence of the ILF, the energy difference between two indicated levels of the first four energy levels for the single quartic AHO and the variation of squared reduced transition matrix elements corresponding to the relevant levels versus the *k*-parameter, as well as the total absorption peaks as a function of the incident photon energy, are given in Figure 5a–c for $\lambda_{1}=0.2$ and $\lambda_{2}=0.5$, respectively. The results of Figure 5c are for k=5nm. As also seen in Figure 5a,b, an increase in the peak amplitudes slightly, and the ILF-induced blue shift at the total absorption peak positions due to the increment in the energy difference between any two energy levels are observed in Figure 5c. The total absorption peak positions of the quartic single AHO potential shift towards higher photon energies than the HO potential considerably.

For $\lambda_{1}<0$$\left(\lambda_{1}=-1\right)$, $\lambda_{2}>0$$\left(\lambda_{2}=0.2\right)$, and k=5nm, the quartic double AHO potential and squared wave functions belonging to the first four levels of the electron confined in this potential in the absence and presence of ILF are given in Figure 6a,b, respectively. As seen in these figures, when there is no ILF, the two wells are decoupled, and the solution to the Schrödinger equation gives rise to a double-fold degenerate ground state with even wave functions for the ground state and odd one for the first excited state. With the effect of ILF, a two-fold degenerate energy level is split. As known, energy splitting plays a primary role in the tunneling phenomenon. As $\lambda_{1}$ increases towards the large negative values, the depths of the wells become deeper and deeper, and quartic double AHO behaves like decoupled two identical single wells for energies lower than the central barrier height. The potential and also energies shift towards higher energies with the effect of ILF, and at the same time, ILF causes a decrease in the central barrier height, and the energies approach the energy values of a single well whose width is increased by two times.

Figure 6c,d show the variations of the first four energy levels as a function of the *k*-parameter for a constant $\lambda_{2}$(=0.2) value and different two values of $\lambda_{1}$, ILF free and ILF induced, respectively. As mentioned earlier, the increase in parameter-*k* causes an increase in the well width and decreases energies in all potential types considered. With the combined effect of *k* and $\lambda_{1}$, the energies lower-lying in the deep and wide well are two-fold degenerate ($E_{1}=E_{2}$); for the higher levels, the degeneracy starts at larger *k*-values. In the shallow wells, degeneracy begins in the weaker confinement regime. In Figure 6e,f, the total absorption peaks as a function of the incident photon energy for quartic double AHO are given in the absence of ILF (solid lines) and the presence of ILF (dashed lines), respectively. Results are for a constant $\lambda_{2}$(=0.2), k=5nm, $\lambda_{1}=-0.5$, and $\lambda_{1}=-1.0$. Total absorption peaks shift to higher photon energies with increasing amplitudes for the shallow quartic double AHO.

It should be kept in mind that the explanations above about the role of both energy differences between the two energy levels for the transitions which we considered and the transition matrix elements on the peak positions and amplitudes of the total ACs are valid in the analysis of the absorption spectrum. In the absence of ILF, the absorption peak belonging to the (1–2) transition is not observed since the first two levels are degenerate (see Figure 6a), and the total AC corresponding to the transition (2–3) is observed at higher incident photon energy than that of the transition (3–4) due to $\Delta E_{43}<\Delta E_{32}$. Degenerate levels split due to the shift towards energies from the central barrier height to higher energies with the effect of ILF, and an absorption peak for related (1–2) transition is also observed (see Figure 6b).

In the absence ($\alpha_{0}=0$-solid lines) and presence of the ILF ($\alpha_{0}=5\;\mathrm{nm}$-dashed lines), energy differences between any two energy levels for both sextic single AHO potential, $V\left(x\right)=V_{0}\left[\lambda_{1}{\left(x/k\right)}^{2}+\lambda_{3}{\left(x/k\right)}^{6}\right]$, for $\lambda_{1}=\lambda_{3}=0.2$, and sextic double AHO potential for $\lambda_{1}=-0.5$, $\lambda_{3}=0.2$ as a function of the width *k*-parameter are given in Figure 7a,b, respectively. The energies for both sextic single and double AHO and, therefore, the difference between any two energy levels mentioned are very sensitive to ILF in the strong confinement regime. In the absence of ILF, the energies are lower because the effective width of the double AHO compared to a single AHO is larger. After ILF is applied, due to the shift of energies from the central barrier height to higher energies, the energy and energy difference have almost the same values as the sextic single AHO. Finally, for the sextic single and double AHOs, ILF free (solid lines) and ILF induced (dashed lines), the total absorption coefficients as a function of the incident photon energy are given in Figure 8a–c. Figure 8a,b belong to the sextic single AHO for $\lambda_{1}=\lambda_{3}=0.2$ and $\lambda_{1}=0.2$, $\lambda_{3}=0.5$, Figure 8c belongs to the sextic double AHO for $\lambda_{1}=-\;0.5$, $\lambda_{3}=0.2$. Results are for k=5nm. As also mentioned before, absorption peaks shift towards the higher photon energies (blue-shift) with increasing amplitudes due to the increasing transition matrix elements with the effect of the ILF. Furthermore, for both sextic single and double AHOs, besides the ILF effect, a blue-shift at the absorption peak positions is observed at large values of $\lambda_{3}$ that causes more strong confinement.

## 4. Conclusions

As known, harmonic and anharmonic oscillators are used to explain the vibrational spectra of diatomic molecules. Due to equal spacing of energy of the HO, all transitions occur at the same frequency, called the single line spectrum. However, many other lines are observed, called overtones, experimentally. Discrepancies between the theoretical and experimental results are resolved by considering the diatomic molecule as an AHO. Furthermore, these potentials play an important role in the quantum tunneling time problems and the spectra of molecules. So, due to the importance of these potentials, we investigated the electronic and optical properties of quantum wells that have harmonic, single, and double AHO potentials by changing the structural parameters and field strength. From our results, it has been seen that the coupling parameters $\left(\lambda_{1},\;\lambda_{2},\;\lambda_{3}\right)$ and ILF have significant effects on the electronic and optical properties of all considered quantum wells. Coupling parameters and ILF cause a significant blue shift in the absorption spectrum.

## Figures and Tables

**Figure 1 nanomaterials-12-01620-f001:**
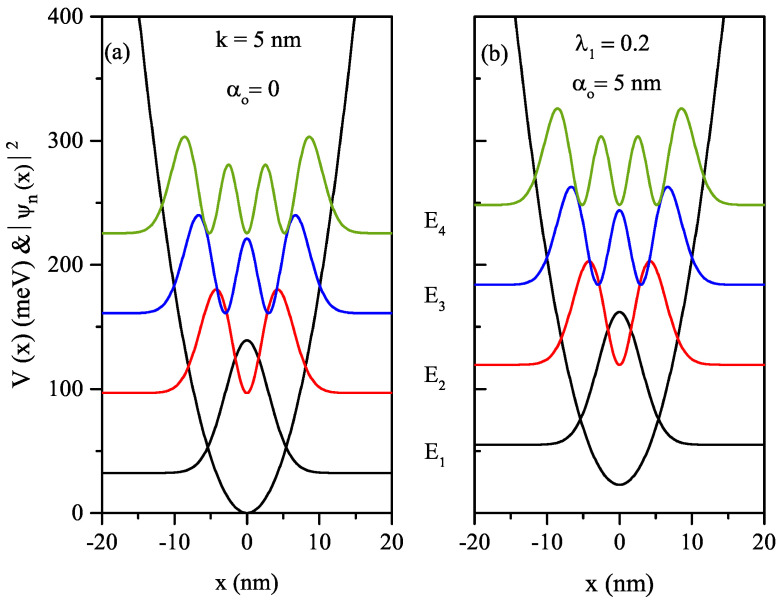
In the absence and presence of the ILF, the HO potential and the squared wave functions of the first four energy levels in relevant energy values in the values of k=5nm and $\lambda_{1}=0.2$ as a function of *x*-growth direction coordinate: $\alpha_{0}=0$ (**a**) and $\alpha_{0}=5\;\mathrm{nm}$ (**b**), respectively.

**Figure 2 nanomaterials-12-01620-f002:**
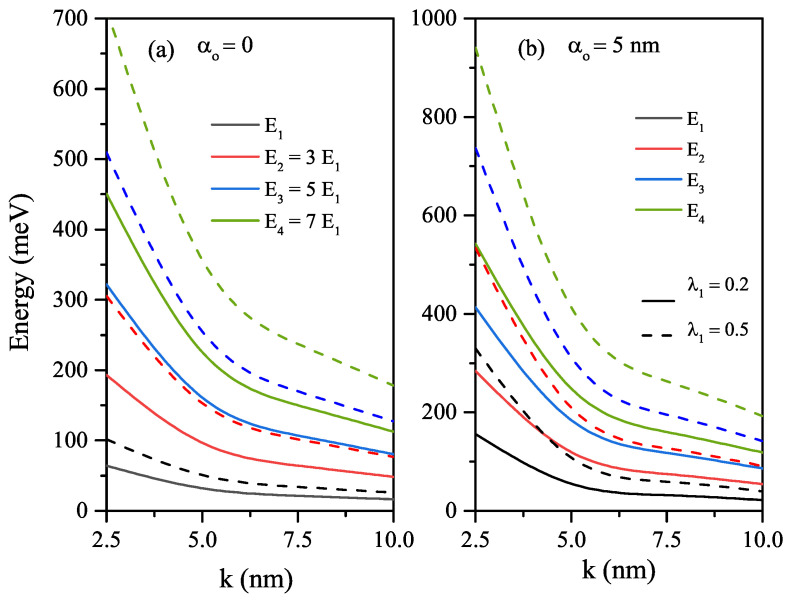
In the absence and presence of the ILF, the variations of the first four energy levels of the HO potential versus the width *k*-parameter for different $\lambda_{1}$ values: $\alpha_{0}=0$ (**a**) and $\alpha_{0}=5\;\mathrm{nm}$ (**b**), respectively.

**Figure 3 nanomaterials-12-01620-f003:**
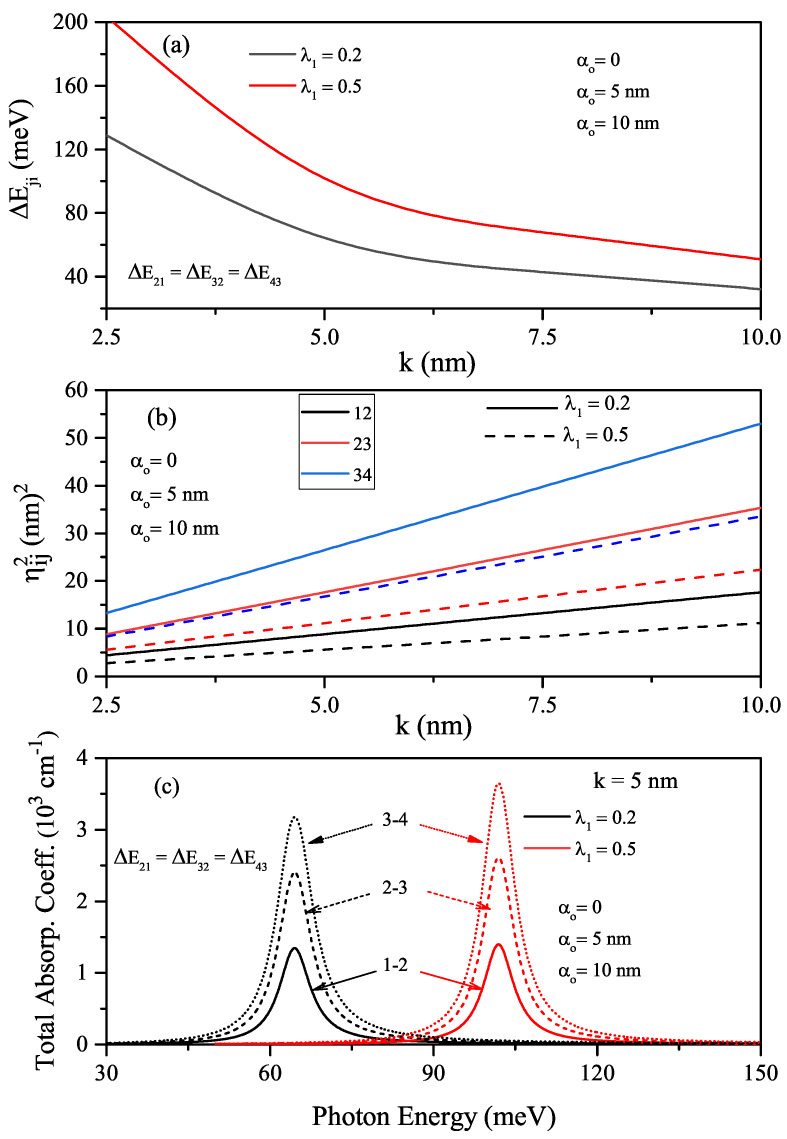
The variations of the energy differences between specified energy levels such as $\Delta E_{21}$, $\Delta E_{32}$, and $\Delta E_{43}$ of the electron confined in the HO potential as a function of the width *k*-parameter (**a**), the squared reduced transition matrix element corresponding to the stated levels as a function of the width parameter (**b**), and the total ACs versus the resonance photon energy for different values of laser-dressed $\alpha_{0}$-parameter and $\lambda_{1}$-parameter (**c**).

**Figure 4 nanomaterials-12-01620-f004:**
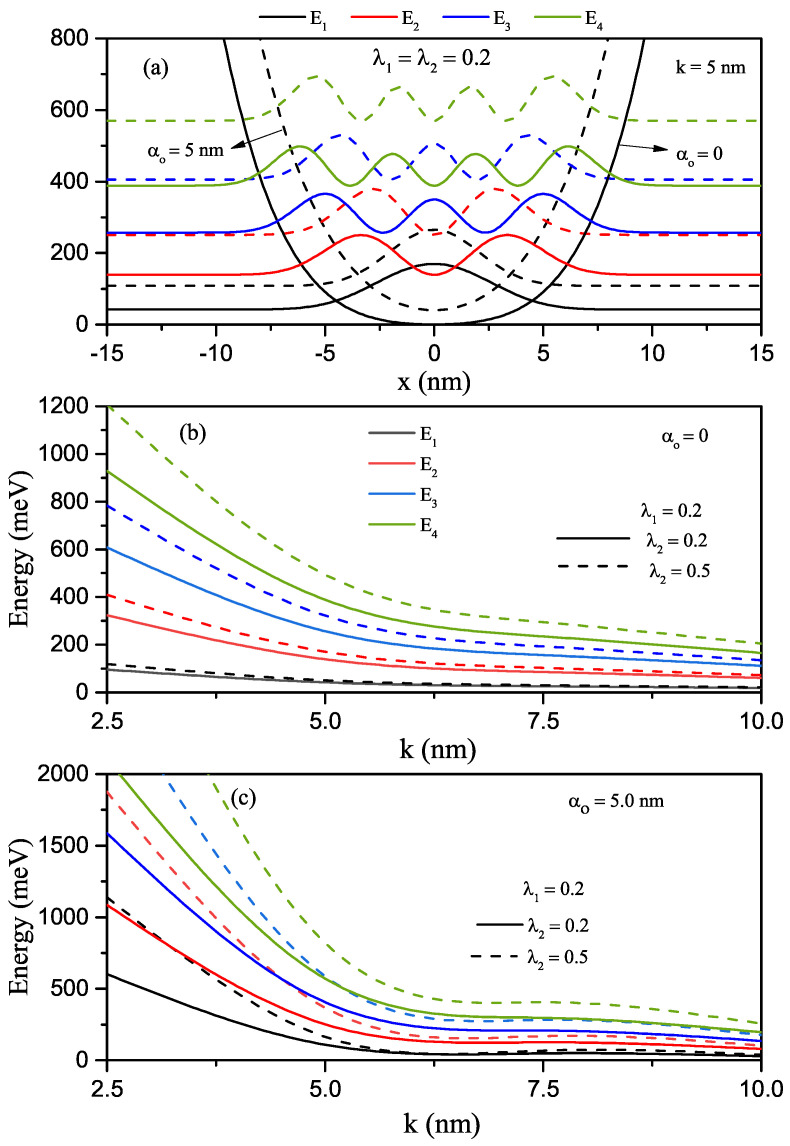
The quartic single AHO potential and squared wave functions of the first four levels in relevant energy values versus the *x*-coordinate for k=5nm and $\lambda_{1}=\lambda_{2}=0.5$ (**a**). Results are for $\alpha_{0}=0$ (solid lines) and $\alpha_{0}=5\;\mathrm{nm}$ (dashed lines). In the absence of ILF, the variations of the first four level energies as a functions of the width *k*-parameter for a constant $\lambda_{1}$ value and two different $\lambda_{2}$ values (**b**). In the presence of ILF, the variations of the first four level energies as a functions of the width *k*-parameter for a constant $\lambda_{1}$ value and two different $\lambda_{2}$ values (**c**).

**Figure 5 nanomaterials-12-01620-f005:**
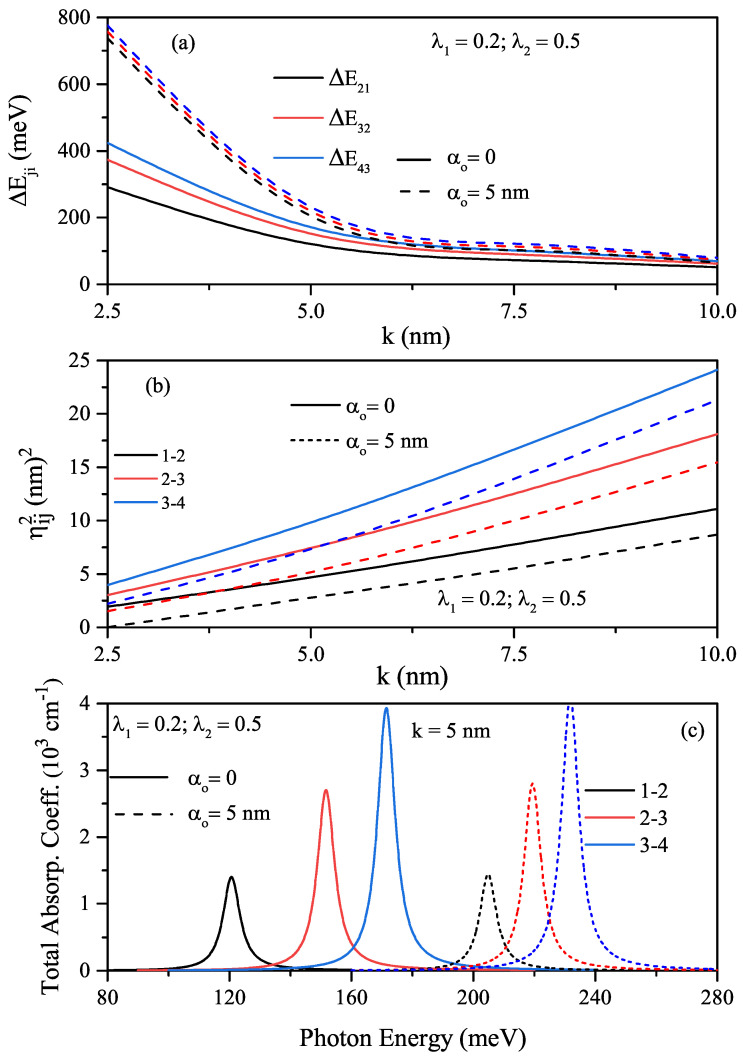
The energy difference between two indicated levels of the electron confined in the quartic single AHO versus the *k*-parameter (**a**), the squared reduced transition matrix elements corresponding to the relevant levels versus the *k*-parameter (**b**), and the variation of total absorption peaks as a function of the incident photon energy for k=5nm (**c**). The results are for $\lambda_{1}=0.2$, $\lambda_{2}=0.5$, $\alpha_{0}=0$ (solid lines) and $\alpha_{0}=5\;\mathrm{nm}$ (dashed lines).

**Figure 6 nanomaterials-12-01620-f006:**
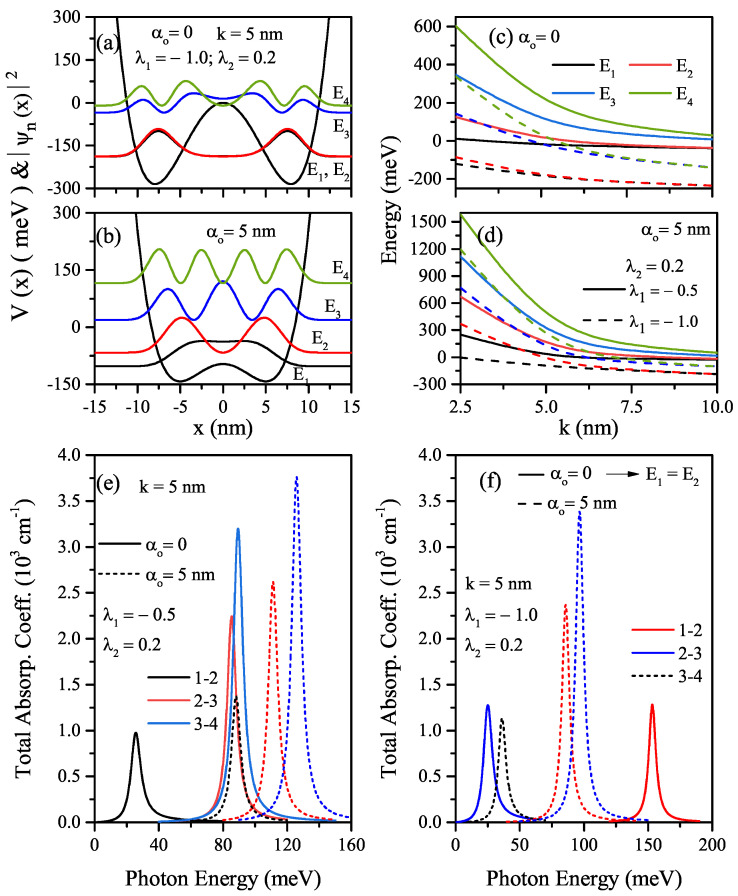
The quartic double AHO potential and squared wave functions of the first four levels in relevant energy values versus the *x*-coordinate for k=5nm, $\lambda_{1}=-1.0$, and $\lambda_{2}=0.2$. Results are for $\alpha_{0}=0$ (**a**) and $\alpha_{0}=5\;\mathrm{nm}$ (**b**). The variations of the first four level energies as a functions of the width *k*-parameter for a constant $\lambda_{2}$ value and two different $\lambda_{1}<0$ values: $\alpha_{0}=0$ (**c**), $\alpha_{0}=5\;\mathrm{nm}$ (**d**), the variation of total absorption peaks as a function of the incident photon energy for k=5nm: $\alpha_{0}=0$ (**e**), $\alpha_{0}=5\;\mathrm{nm}$ (**f**).

**Figure 7 nanomaterials-12-01620-f007:**
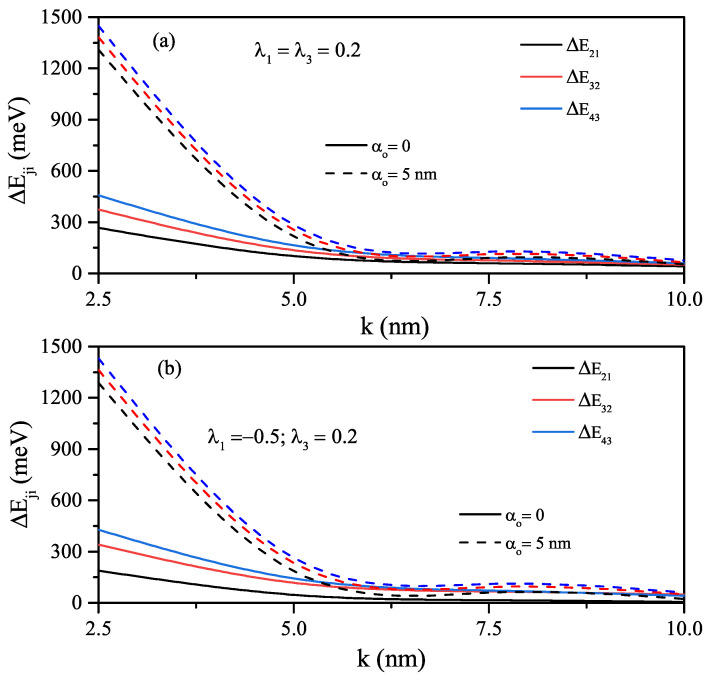
The variations of the energy differences between specified energy levels such as $\Delta E_{21}$, $\Delta E_{32}$ and $\Delta E_{43}$ as a function of the width *k*-parameter: for the electron confined in the sextic single AHO potential $\left(\lambda_{1}=\lambda_{3}=0.2\right)$ (**a**), and for the electron confined in the sextic double AHO potential $\left(\lambda_{1}=-0.5,\lambda_{3}=0.2\right)$ (**b**). Results are for $\alpha_{0}=0$-solid lines and $\alpha_{0}=5\;\mathrm{nm}$-dashed lines.

**Figure 8 nanomaterials-12-01620-f008:**
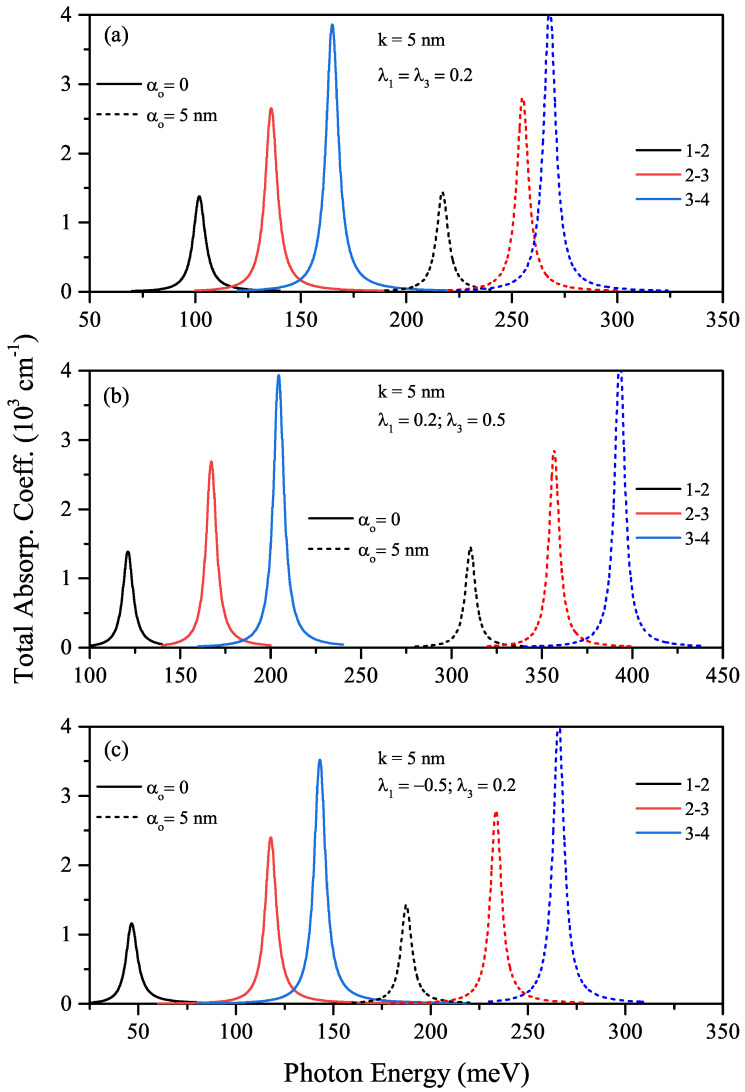
The variations of total absorption coefficients as a function of the incident photon energy for k=5nm: the sextic single AHO for $\lambda_{1}=\lambda_{3}=0.2$ (**a**), $\lambda_{1}=0.2,\lambda_{3}=0.5$ (**b**), and to the sextic double AHO for $\lambda_{1}=-\;0.5,\lambda_{3}=0.2$ (**c**), respectively.

## Data Availability

No new data were created or analyzed in this study. Data sharing is not applicable to this article.

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
