# Peer review of "Effects of Intense Laser Field on Electronic and Optical Properties of Harmonic and Variable Degree Anharmonic Oscillators"

_nanomaterials, 2022, doi:10.3390/nano12101620_

Round 1
Reviewer 1 Report
In this manuscript, the authors calculated the electronic and optical properties of the harmonic oscillator and single and double anharmonic oscillators including higher-order anharmonic terms under the non-resonant intense laser field. In my opinion, this manuscript is interesting to the readers of Nanomaterials. The topic is very important in this field. This work is novel and original. The authors have solid background in this field. Therefore, the referee recommends it to be published after the following revisions:
1. The English should be polished by a native speaker.
2. Please use the same style to prepare the figures (e.g. Fig.3). The frame thickness is different for Fig. 3(a) and Fig. 3(b).
- Please cite more recent reports (2019~2022).
In general, this work seems to be very interesting. The referee would like to see the revision if possible.

Author Response
Referee 1:
The Referee:
In this manuscript, the authors calculated the electronic and optical properties of the harmonic oscillator and single and double anharmonic oscillators including higher-order anharmonic terms under the non-resonant intense laser field. In my opinion, this manuscript is interesting to the readers of Nanomaterials. The topic is very important in this field. This work is novel and original. The authors have solid background in this field. Therefore, the referee recommends it to be published after the following revisions:
Our answer
We want to thank and express our gratitude to the Referee for his/her excellent report, which we believe has helped us substantially improve the quality and clarity of our manuscript.
The Referee:
- The English should be polished by a native speaker.
Our answer
We thank the Referee for his/her suggestion. The English has been revisited and polished by a native speaker.
The Referee:
- Please use the same style to prepare the figures (e.g. Fig.3). The frame thickness is different for Fig. 3(a) and Fig. 3(b).
Our answer
We thank the Referee for his/her suggestion. The figures have been homogenized with the same style
The Referee:
- Please cite more recent reports (2019~2022).
Our answer
We thank the Referee for his/her suggestion. In the second paragraph of the revised versión we added thw following text with the corresponding references:
“The transport of heat along a chain of particles interacting through anharmonic potentials consisting of quartic terms in addition to harmonic quadratic terms and subject to heat reservoirs at its ends has been reported by Landi and Oliveira showing that the introduction of the energy conserving stochastic noise leads to Fourier’s law \cite{Ref1}. In the theoretical work of N. H. Fletcher it is concluded that the term "inharmonic" is an appropriate descriptor for classical oscillators, such as metal bars, that have nonharmonic vibrational spectra even in the linear limit of small vibrations \cite{Ref2}. By using an exact and perturbative form, the repulsive harmonic oscillator and an extension of this system, with an additional linear anharmonicity term, has been studied \cite{Ref3}. The authors compared the perturbative solution up to second order with the exact solution when the system is initially in coherent and a Schr\"odinger-cat states. Villegas-Mart\'inez and co-workers proposed an algorithm for the localization of normal modes in suitable subsets that are chosen to strictly limit the errors introduced by the harmonic couplings while still leading to maximally localized modes \cite{Ref4}. By a suitable choice of coordinates and by using their proposed algorithm, the computational effort required for calculations of anharmonic vibrational spectra can be reduced significantly. Sharma and Sastri reported the numerical solution of the quantum anharmonic oscillator by using matrix diagonalization technique \cite{Ref5}. In their work, where the interaction potential consists of quadratic and quartic terms, it was embedded within an infinite square well potential of appropriate width and its sine eigen functions were used as basis functions for the employed diagonalization matrix method.”
The Referee:
In general, this work seems to be very interesting. The referee would like to see the revision if possible.
Our answer:
Finally, we hope that the Referee finds our responses to his/her comments, questions, and suggestions satisfactory and that he/she considers the revised version of our manuscript suitable for publication in the Nanomaterials Journal.

Reviewer 2 Report
Authors present simulation results of the harmonic and anharmonic properties. Manuscript is well written and logically structured. All figures are neat and well detailed -same applies to the description. I have no issue with the references.
I cannot confirm mathematical soundness as this would obviously require an access to the simulations, however, that's basically the nature of reviewing theoretical and numerical simulation results.
Some typo mistakes to be corrected:
-line 49 (after citation)
-line 57 (citation)
I would like to thank authors for an effort, it's always a pleasure to read well written manuscripts. I wish you further success.
All the best.
Author Response
Referee 2:
The Referee:
Authors present simulation results of the harmonic and anharmonic properties. Manuscript is well written and logically structured. All figures are neat and well detailed -same applies to the description. I have no issue with the references.
I cannot confirm mathematical soundness as this would obviously require an access to the simulations, however, that's basically the nature of reviewing theoretical and numerical simulation results.
Our answer
We want to thank and express our gratitude to the Referee for his/her excellent report, which we believe has helped us substantially improve the quality and clarity of our manuscript.
The Referee:
Some typo mistakes to be corrected:
-line 49 (after citation)
-line 57 (citation)
I would like to thank authors for an effort, it's always a pleasure to read well written manuscripts. I wish you further success.
Our answer
We want to thak the Referee for his/her very positive opinion about our mansuscript. The typos have been amended.
Finally, we hope that the Referee finds our responses to his/her comments, questions, and suggestions satisfactory and that he/she considers the revised version of our manuscript suitable for publication in the Nanomaterials Journal.